Assessing morphological variations in the seagrass genus Halodule (Cymodoceaceae) along the Brazilian coast through genetic analyses

de Sousa Virgínia Eduarda 1 2 vihobb25@gmail.com
da Silva Cortinhas Maria Cristina F. 2
http://orcid.org/0000-0002-1722-0806 Creed Joel C. 3
Batista Maria Gardênia S. 4
http://orcid.org/0000-0002-5504-5770 Proietti Maira C. 2
http://orcid.org/0000-0001-5771-829X Copertino Margareth 1 marga.copertino@gmail.com
1 Programa de Pós-Graduação em Oceanografia Biológica, Instituto de Oceanografia, Universidade Federal do Rio Grande , Rio Grande, Rio Grande do Sul , Brazil
2 Laboratório de Ecologia Molecular Marinha, Instituto de Oceanografia, Universidade Federal do Rio Grande , Rio Grande, Rio Grande do Sul , Brazil
3 Departamento de Ecologia, Universidade do Estado do Rio de Janeiro , Rio de Janeiro, Rio de Janeiro , Brazil
4 Centro de Ciências da Natureza/Biologia, Universidade Estadual do Piauí , Teresina, Piauí , Brazil
Mikami Koji
Electronic publication date: 2025 Mar 19
Publication date: 2025
Volume: 13
Electronic Location ID: e19038
Received 2023 Oct 31; Accepted 2025 Jan 31
Copyright: © 2025 Sousa et al.
Copyright year: 2025
Copyright holder: Sousa et al.
License: This is an open access article distributed under the terms of the Creative Commons Attribution License, which permits unrestricted use, distribution, reproduction and adaptation in any medium and for any purpose provided that it is properly attributed. For attribution, the original author(s), title, publication source (PeerJ) and either DOI or URL of the article must be cited.
License URL: https://creativecommons.org/licenses/by/4.0/

Keywords: Phylogenetic, DNA barcoding, Marine angiosperm, Seagrasses, Atlantic, Morphology, Taxonomy

Funding: Conselho Nacional de Desenvolvimento Científico e Tecnológico (CNPq) 442206/2020-8 CAPES—PRINT—Process 88881.310827/2018-01 Programa de Internacionalização da Fundação Coordenação de Aperfeiçoamento de Pessoal de Nível Superior This work was supported by the Conselho Nacional de Desenvolvimento Científico e Tecnológico (CNPq) (Process No. 442206/2020-8). The APC was funded by CAPES—PRINT—Process—88881.310827/2018-01, “Programa de Internacionalização da Fundação Coordenação de Aperfeiçoamento de Pessoal de Nível Superior”. The funders had no role in study design, data collection and analysis, decision to publish, or preparation of the manuscript.

==============================
Background

Seagrass meadows are distributed globally and provide critical ecological functions and ecosystem services, but seagrasses are still poorly studied compared with other coastal and marine foundation species. Species taxonomy is uncertain in various seagrass genera, such as the genus Halodule. Until recently, the morphological characteristics of leaves were the major criteria for species identification. In Brazil, three species of Halodule are recognized and separated solely using leaf morphology criteria by some authors; however, the leaves present high variability and plasticity, resulting in great uncertainty about species diversity. A review of seagrass species validation using both morphological and phylogenetic methods is needed. This includes examining the genus Halodule with the aim of better understanding its diversity and spatial distribution and consequently supporting management and conservation goals.

Methods

Plant samples with the morphological forms of H. beaudettei and H. wrightii were collected at five sites across three Brazilian marine ecoregions. Leaf tip format and leaf width and length were compared among all the sites and between the two populations with different leaf tip forms. Molecular diversity and divergence indices and analyses were used to estimate the genetic distance between H. wrightii and H. beaudettei populations. To determine the phylogenetic relationship between the two morphologies, we sequenced two molecular markers, the internal transcribed spacer (ITS) fragment and the rbcL gene, to construct phylogenetic trees using Bayesian inference.

Results

We identified H. beaudettei morphology at two sites in Northeast Brazil, while H. wrightii was found in all the ecoregions in the remaining areas, distinguished by the leaf tip shape that occurred at each site. Leaf width and length varied across the five sites, and leaf length differed between H. wrightii and H. beaudettei, with higher values observed in H. beaudettei. Variations in morphological measurements may be related to habitat conditions at each site studied. No divergence was observed for the DNA sequences of two molecular markers, except for a single base in the ITS region, resulting in the Brazilian specimens merging at a single node in the phylogenetic trees. AMOVA and genetic distance analysis revealed low genetic variation but high structuring within populations. The ITS marker showed insufficient genetic variance to delineate the two morphologies as different species which indicating H. wrightii and H. beaudettei are closely related. A genomic approach is needed to fully resolve this issue. This study represents the first step toward the complete determination of the Halodule genus on the Brazilian coast.

Introduction

Seagrasses are a unique group of vascular plants completely adapted to submerged marine and coastal environments, and they provide numerous ecosystem services (Costanza et al., 1997; Nordlund et al., 2016), such as increasing biodiversity, supporting fisheries (Boström, Jackson & Simenstad, 2006), protecting coastal areas (Koch et al., 2009), cleaning water (Short & Short, 1984; Sanchez-Vidal et al., 2021) and sequestering carbon (Mcleod et al., 2011). Moving into the marine environment relatively recently (70–100 million years ago) compared to algae, seagrasses are characterized by low taxonomic diversity (~72 species within 14 genera) and high adaptive convergence in terms of morphological traits (Short et al., 2007, 2011; Wissler et al., 2011). Until recently, leaf morphology and anatomy have been the primary attributes used for taxonomic studies and species identification in seagrasses (den Hartog, 1964; Phillips, 1967). However, their high phenotypic plasticity, an ability shared by most plant groups, scarcity of sexual reproductive parts and lack of complementary genetic analyses have hindered taxonomic studies, resulting in the underestimation or overestimation of the number of species in some genera (Spalding et al., 2003; Waycott et al., 2006; Bricker et al., 2011; McDonald et al., 2016). Since the end of the 20th century, seagrass species have undergone phylogenetic reassessment using genetic markers to supplement morphological and anatomical studies.

The seagrass genus Halodule is widely distributed along tropical and warm-temperate coasts and is present in five (of a total of six) global seagrass bioregions: the temperate North Atlantic, tropical Atlantic, Mediterranean, temperate North Pacific and Tropical Indo-Pacific bioregions (Short et al., 2007). Halodule wrightii occurs in the tropical and subtropical Atlantic Ocean, Caribbean, Gulf of Mexico and tropical eastern Pacific (Phillips & Meñez, 1988). It is a relatively short-lived, polyploid pioneer species that colonizes disturbed habitats and can tolerate a wide range of temperatures, submergence depths, and salinities, preferring high salinity over low salinity (Creed, 2003; Rivera-Guzman et al., 2017).

Along the southwestern Atlantic coast of Brazil, three species of Halodule have been mentioned by different authors: Halodule wrightii (HW), with the broadest distribution in the tropics and subtropics (4°S; 42°W to 27S; 48°W); H. emarginata from the semiarid northeastern region to southeastern Brazil (24°S; 46°W to 25°S; 48°W) (den Hartog, 1964; Oliveira, Pirani & Giuletti, 1983; Copertino et al., 2016); and H. beaudettei (HB), which is confined to estuaries from Piauí and Ceará states along the semiarid coast of Northeast Brazil (Magalhães & Barros, 2017). Traditionally, the last two assumed species can be distinguished from H. wrightii by their leaf morphology, mainly leaf length and width and the leaf tip format, which can be triangular (H. wrightii), tridentate (H. beaudettei) or notched/rounded (H. emarginata) (den Hartog, 1964; Oliveira, Pirani & Giuletti, 1983) (Table S1). In general, the reproductive parts of Halodule are rarely observed in meadows (Ito & Tanaka, 2011).

Since early seagrass taxonomic descriptions for the western Atlantic, H. beaudettei has been described as a distinct species (den Hartog, 1967) or as a synonym (Phillips, 1967; Eiseman, 1980). After several samples from Florida were compared, Phillips (1967) reported that the characteristics of H. beaudettei and H. wrightii are connected by a complete range of intermediate morphological forms. More recently, the authors described the H. beaudettei leaf tip trait in Mexico (van Tussenbroek et al., 2010), Caribbean (Martínez-Daranas, 2002), and Principe Island in the South Atlantic (Alexandre et al., 2017), and this trait is very similar to what has been reported in Northeast Brazil (Magalhães & Barros, 2017). Since leaf characteristics can vary with age, environmental conditions and functional traits, the validity of H. beaudettei as a separate species is still a pertinent issue and requires clarification through complementary morphological and phylogenetic approaches (Wheeler, Furman & Hall, 2020; Moreira-Saporiti et al., 2023).

Determining the status of seagrass species on the basis of only the morphology of vegetative parts can hamper the knowledge of the actual status of biodiversity and distribution. For example, H. beaudettei and H. emarginata are listed as data deficient in the International Union for Conservation of Nature (IUCN) Red List of Threatened Species category, perhaps resulting from the lack of consensus of their status as true species, whereas H. wrightii is categorized as Least Concern. Conservation strategies and efforts rely on the continuous mapping of species populations, so there is demand for identifying their diversity and inferring their range distribution to achieve the goal of successfully monitoring each species. In addition, the correct delineation of species is helpful in measuring ecosystem biodiversity and determining applicable identification techniques, facilitating their recognition in the field by researchers and citizens.

The application of phylogenetic analysis is an effective tool for species confirmation (Dussex et al., 2018; Lin et al., 2021) and resolution of genus taxonomy (Petersen et al., 2014; Singh, Southgate & Lal, 2019; Liu, Kumara & Hsu, 2020). Sequencing genomic regions via the DNA barcoding technique allows the identification and recognition of new species (Hebert et al., 2003; Waycott et al., 2006). In plants, several markers, such as the genes phyB, rbcL and matK and the intergenic fragments ITS and trnH-psbA (CBOL Plant Working Group et al., 2009; Yao et al., 2010), are used for phylogenetic study. The fragments along the DNA used as markers can be considered as polymorphic or conserved (Liu, Kumara & Hsu, 2020; Lin et al., 2021; Kwan et al., 2023), thus, integrating different markers is convenient for phylogenetic resolution.

Previous studies have investigated the phylogeny of H. wrightii and H. beaudettei and showed contrasting results. Ito & Tanaka (2011) used the rbcL marker and found a high genetic similarity between samples of H. wrightii and H. beaudettei in the Pacific; while a sequence named H. beaudettei from the rbcL marker is available in the GenBank database and is distinct from other H. wrightii sequences (Les, Cleland & Waycott, 1997; Wagey & Calumpong, 2013). Hence, the species status of H. beaudettei morphology is still unclear for Brazil and other regions along the Atlantic coast (Samper-Villarreal et al., 2018). Given the current uncertainties, this study aims to comparatively evaluate the morphological variations in Halodule specimens found across the Brazilian coast through morphological and genetic analysis. The genetic distance between the two morphologies was assessed by combining the polymorphic ITS and conserved rbcL DNA regions to survey the degrees of similarity in the Brazilian samples. We expect to advance in terms of clarifying the genetic characteristics of Halodule in Brazil and highlighting the morphological forms observed in the samples.

Materials and Methods

Study site

The study area spans 4,800 km of the Brazilian coastline between latitudes 4°S and 28°S and includes three marine ecological regions (sensu Spalding et al., 2007): northeastern Brazil (NB), eastern Brazil (EB) and southeastern Brazil (SEB) (Fig. 1). The sites in each ecoregion were NB: Praia de Macapá (MAC), Timonha Ubatuba estuarine system (SETU) and Praia da Pedra do Sal (PS) (two estuaries and a sandy beach, respectively) all in Piauí state; EB: Praia de Manguinhos (PM) in Búzios (at a 2-m deep sandy beach) in Rio de Janeiro State; and SEB: Lagoa da Conceição (LC) (coastal lagoon) in Santa Catarina state (Fig. 1).

Figure 1 Map of sampled sites along the Brazilian coast.

PS, Praia da Pedra do Sal; MAC, Praia de Macapá; SETU, Timonha-Ubatuba estuarine system; PM, Praia de Manguinhos; LC, Lagoa da Conceição. Base maps source: https://www.ibge.gov.br/geociencias/organizacao-do-territorio/malhas-territoriais/15774-malhas.html[p].

The sampling area in Piauí state (NB ecoregion) is on a semiarid coast at latitude 4°S and has well-defined wet and dry seasons, with maximum precipitation occurring during the austral autumn (Nimer, 1989). Located in the equatorial zone, the climate is hot and humid, with temperatures between 22 °C and 31 °C (Mai & Rosa, 2009). Praia de Manguinhos in Armação de Búzios in northern Rio de Janeiro state (EB ecoregion), latitude 22°S, is an embayed beach located on the northern side of Cape Buzios, which is wave protected (Bulhoes & Fernandez, 2011), and the climate is hot and humid, with a maximum rainfall rate during the austral summer (Muehe & Lins-de-Barros, 2016). Lagoa da Conceição, on Florianópolis Island in Santa Catarina state (SEB ecoregion, latitude 27°S), is in the warm temperate zone and is a coastal lagoon with a narrow channel connecting to the sea (Dominguez, 2004). The average temperature at Lagoa da Conceição is between 16 °C and 25 °C, and the rainy season is in the austral summer (Beduschi, 2010).

The sampling meadows in each ecoregion were selected on the basis of previous studies of the distribution of Halodule in Brazil (Creed, 2003; Magalhães & Barros, 2017; Silva et al., 2018; Bercovich et al., 2019). We identified the plants by examining the leaf tips, following den Hartog (1967) and Magalhães & Barros (2017). We collected the samples randomly and manually along the meadow, maintaining a minimal distance of 2 m between samples, and these samples were used for both molecular and morphological analyses. As modular individuals, each sample was composed of several shoots with associated leaves and rhizomes. In the field, the samples were cleaned with seawater, transported in a cooler and then cleaned in distilled water with scissors and a brush being used to remove salt and debris. The cleaned plants were stored in Ziplock bags with silica gel for drying. The samples, each containing three or more leaves, were stored in a −80 °C freezer for DNA analysis or in 70% alcohol for morphological analysis. Field collection was performed under license No 45819-1, which was provided by Sistema de Autorização e Informação em Biodiversidade (SISBIO).

Morphological analysis

For morphological analysis, a total of 50 plants (10 samples from each study site) were portioned and maintained in 70% alcohol until dry and pressed into herbarium sheets (Table S1). The plant leaf tips were observed, recorded and photographed via optical microscopy (Leica MZ9.5). The length and width of the leaves were obtained with a millimeter steel ruler (Trident). Differences in morphometric parameters between the five sites were compared via the Kruskal‒Wallis test and Dunn’s post hoc test, and differences in morphometric parameters between the two leaf tip shape samples were compared via the Mann‒Whitney U test, since none of the results met the homoscedasticity criterion in the statistical environment R 4.2.1 (R Core Team, 2021), with the Rstatix (Kassambara, 2023), dplyr (Wickham et al., 2023) and DescTools (Signorell, 2024) packages.

DNA extraction and amplification

In total, 30 leaf samples were selected for DNA extraction (Table S1), which followed a modified 2x CTAB protocol (Information S1) (Doyle & Doyle, 1987). We amplified two DNA regions that were recommended by the Consortium for the Barcode of Life (CBOL) and that had been used in previous studies (Kato et al., 2003; Lin et al., 2021; Nguyen et al., 2021; Petersen et al., 2014; Wagey & Calumpong, 2013) for species identification in plants: the internal transcribed spacer (ITS) of the ribosomal DNA, which uses the primers P674 and P675 described by Nguyen et al. (2015), and the rbcL gene in the chloroplast, which uses the forward primer P610 described by Kress & Erickson (2007) and the reverse primer HAL986 that consists of 986 base pairs (Table 1). The reverse primer HAL986 was designed at the Marine Molecular Ecology Laboratory of the Rio Grande Federal University and uses the partial rbcL gene of the chloroplast from the species Halodule wrightii (GenBank HQ901575 and A571197), Halodule uninervis (GenBank AY952436 and KP739815) and Halodule pinifolia (GenBank KF488493) as a model. An OligoAnalyzer (PrimerQuest program, accessed 25/02/2022) was used to verify the efficiency of the primers.

Table 1 Primers and sequences for the locus analysed.

Locus	Primer	Primer sequence	Reference	
ITS	P674	5′ CCTTATCATTTAGAGGAAGGAG 3′	Nguyen et al. (2015)	
	P675	5′ TCCTCCGCTTATTGATATGC 3′	
rbcL	P610	5′ ATGTCACCACAAACAGAGACTAAA 3′	Kress & Erickson (2007)	
	HAL986	5′ CCAGCGTGAATATCATCTCCACC 3′	This study	

The PCR mixtures in 25 μl reactions for the ITS and rbcL fragments followed Nguyen et al. (2015) and Lucas, Thangaradjou & Papenbrock (2012), respectively, with modifications, which included 1x buffer, 2 mM MgCl2, 0.2 mM dNTPs, 1 pmol of each primer, 1 U Taq Polymerase, 0.02% BSA, and 10–30 ng of DNA. The PCRs were performed in a gradient thermocycler (Veriti 96; Thermo Fisher Scientific, Waltham, MA, USA) under the following conditions: for ITS, initial denaturation for 4 min/95 °C, followed by 30 cycles of denaturation (25 s/95 °C), primer annealing (30 s/54 °C) and extension (35 s/72 °C), and a final extension for 3 min/72 °C; for rbcL, initial denaturation for 5 min/95 °C, followed by 30 cycles of denaturation (30 s/94 °C), primer annealing (30 s/60 °C), extension (1 min/70 °C), and a final extension for 8 min/72 °C. The amplified products were stained with GelRed (0.001%) and subjected to electrophoresis on 1% agarose gels immersed in TBE buffer. A molecular size marker (1 kb) was used to estimate the total length of the fragments. The gels were visualized on a UV Transilluminator UVPR M20 and then photographed. DNA purity and concentration were measured with a Biodrop® 2000 Spectrophotometer.

The PCR products were purified to 30 μl with a purification kit (MEBEP Bio Science), and the forward and reverse strands were sequenced via capillary electrophoresis (ACTGene; Porto Alegre, Brazil).

Data analyses

Each sequence was evaluated with the BLAST tool in the National Center for Biotechnology Information (NCBI) GenBank, and the sequences were then aligned with the ClustalW multiple alignment method (Thompson, Higgins & Gibson, 1994) and cut manually to 648 bp (ITS) and 836 bp (rbcL) with BioEdit 7.2.5 (Hall, 1999). The sequences were deposited in GenBank under the accession numbers OR284885–OR284889 for the ITS fragment and OR345358–OR345362 for the rbcL gene.

We constructed a haplotype network to visualize the number and distribution of haplotypes across the five sampled areas with packages pegas (Paradis, 2010), ggspatial (Dunnington, 2023) and ggplot2 (Wickham, 2016) in R. DNAsp (Rozas et al., 2017) was used to calculate the number of haplotype (H), haplotype (h) and nucleotide (π) diversities and the number of polymorphic sites (S). Arlequin 3.5.2.2 (Excoffier, Laval & Schneider, 2005) was used to calculate the molecular variance (AMOVA) to describe the genetic variation in two instances, one comparing the five sampled populations and one comparing the study sites in two regions: the northeast (PS, MAC and SETU) and the south (PM and LC). Standard AMOVA was performed by applying F-statistics with 10,000 permutations due to the smaller number of samples. Fst pairwise differences were also calculated in Arlequin to assess the genetic differentiation between the five populations assuming the haplotype frequency and 10,000 permutations. To estimate the genetic distance between the populations with H. wrightii and H. beaudettei leaf tips and the haplotypes obtained from the populations, we calculated the Δst (Nei, 1982) index in DNAsp and the genetic distance values in MEGA11 (Tamura, Stecher & Kumar, 2021), with 10,000 bootstrap replications, a gamma parameter of 0.50 and a Tajima‒Nei model, for the ITS sequences.

To determine the phylogenetic affinity of the Brazilian Halodule samples, we constructed phylogenetic trees via the Bayesian inference (BI) method. Bayesian inference was estimated via Mrbayes 3.2.7 (Ronquist & Huelsenbeck, 2003) according to the following settings: two runs with four simultaneous chains with 1 million generations, a burn-in rate of 25%, and default prior information. The analysis considered the most suitable evolutionary models, as defined in JModelTest 2.1.9 by the lowest AIC (Akaike information criterion) values: HKY+G for ITS and K80+I for rbcL (Guindon & Gascuel, 2003; Darriba et al., 2012). Additional sequences from GenBank were incorporated into the trees: H. uninervis and H. pinifolia for both markers; other H. wrightii sequences; H. beaudettei for rbcL; and a sequence of H. emarginata from Brazil for ITS. The species Cymodocea nodosa was selected as the outgroup.

Results

Leaf morphology

In terms of the shape of the leaf tips, the form of H. beaudettei, which has a tridentate apex with a median tooth longer than the two lateral teeth, was found only in the estuarine regions of the Macapá River (MAC) and Timonha-Ubatuba (SETU), Piauí state, NB ecoregion. The remaining sites (PS, PM and LC) included the H. wrightii form—marginal triangular teeth with concave inner surfaces and median teeth that are usually shorter (Fig. 2, Table S1). Halodule emarginata leaf tips were not detected at the sample sites. We observed discrepancies in the measured values between the two leaf tip forms of the HW and HB populations in terms of leaf length (P = 0.0001772) but no difference in leaf width (P = 0.802) (Fig. 3). Morphometric measurements among the five sites revealed significant differences in leaf length (P = 0.0009349) and leaf width (P = 0.01853) (Fig. 4).

Figure 2 Variability of leaf tips collected across sampled areas.

(A) H. beaudettei in Timonha-Ubatuba estuarine system; (B) H. beaudettei in Praia de Macapá; (C) H. wrightii in Praia de Manguinhos; (D) H. wrightii in Lagoa da Conceição.

Figure 3 Morphological measures of samples between the leaf tip forms of H. wrighii and H. beaudettei with ± standard error: leaves length (A) and width (B).

HB, H. beaudettei; HW, H. wrightii.

Figure 4 Morphological measures of Halodule samples across the surveyed sites with ± standard error: leaves length (A) and width (B).

PS, Praia da Pedra do Sal; MAC, Praia de Macapá; SETU, Timonha-Ubatuba estuarine system; LC, Lagoa da Conceição; PM, Praia de Manguinhos.

BLAST analysis

In BLAST, the rbcL sequences from all sites matched the previous H. wrightii sequences. The ITS sequences were associated with H. emarginata and H. uninervis since there are no H. wrightii ITS sequences available on the platform; moreover, we did not observe the presence of named H. beaudettei sequences for the internal transcribed spacer (ITS) molecular marker.

Phylogenetic analysis

For the ITS and rbcL region fragment analysis, 30 Halodule samples were used: 16 from Piauí, six from Santa Catarina and 10 from Rio de Janeiro. Two haplotypes and one polymorphic site (parsimony informative site) were observed in the ITS fragment, the first haplotype was found in the population of Piauí (PS and SETU), whereas the populations of Santa Catarina, Rio de Janeiro and Piauí (MAC) shared the second haplotype with low haplotypic (h) and nucleotide diversity (π) (Fig. 5, Table 2, Table S1). The AMOVA that was performed considering the two groups separated by the regions (northeast and south) and by the five sampled populations (Table S1) resulted in values of 0.00 (P = 1.00) within (ϕST) and among (ϕSC) populations and among groups (ϕCT), with 100.00% variation within populations for both analyses (Table S2). The Fst pairwise difference across the five sampled populations was 0.00 (Table 3). The Δst index was 0.00003, for the genetic difference between the H. wrightii and H. beaudettei populations, and 0.00065 for the two haplotypes. Low genetic distance were obtained when the five populations were compared (Table 4) and between the two haplotypes (0.00155 ± 0.00157) and the HB and HW leaf tip populations (0.00077 ± 0.00077). The rbcL region presented only one haplotype shared by all 30 sequences, and the haplotype diversity (h), nucleotide diversity (π), and number of polymorphic sites (S) were 0.00. The percentage of sequence variation (AMOVA) was 0.00 (P = 1.00) within and among the populations examined.

Figure 5 Haplotype network and distribution of each haplotype across the surveyed sites.

Table 2 Diversity of the ITS region (648 pb) for Halodule from the three sampled Brazilian states and for the two leaf tip format.

State	N	H	S	h	π	
Piauí	14	2	1	0.4945+/−0.0876	0.0007+/−0.0007	
Santa Catarina	6	1	0	0.0000+/−0.0000	0.0000 +/−0.0000	
Rio de Janeiro	10	1	0	0.0000+/−0.0000	0.0000 +/−0.0000	
Leaf tip						
H. beaudettei	9	2	1	0.5556+/−0.0765	0.0009+/−0.0009	
H. wrightii	21	2	1	0.38095+/−0.0851	0.0006+/−0.0006	
Note:

N, number of individuals; H, number of haplotypes; S, number of polymorphic sites; h, haplotype diversity; π, nucleotide diversity.

Table 3 Pairwise FST differentiation (below diagonal) of the ITS fragment (648 pb) for Halodule across the five study sites and corrected p-values (above diagonal).

	SETU	MAC	PS	PM	LC	
SETU	0.00000	0.99990 ± 0.00	0.99990 ± 0.00	0.99990 ± 0.00	0.99990 ± 0.00	
MAC	0.00000	0.00000	0.99990 ± 0.00	0.99990 ± 0.00	0.99990 ± 0.00	
PS	0.00000	0.00000	0.00000	0.99990 ± 0.00	0.99990 ± 0.00	
PM	0.00000	0.00000	0.00000	0.00000	0.99990 ± 0.00	
LC	0.00000	0.00000	0.00000	0.00000	0.00000	
Note:

SETU, Timonha-Ubatuba estuarine system; MAC, Praia de Macapá; PS, Praia da Pedra do Sal; PM, Praia de Manguinhos; LC, Lagoa da Conceição.

Table 4 Genetic distance (below diagonal) of the ITS fragment (648 pb) for Halodule across the five study sites and standard error (above diagonal).

	PS	MAC	SETU	PM	LC	
PS		0.00153	0.00000	0.00153	0.00153	
MAC	0.00154		0.00153	0.00000	0.00000	
SETU	0.00000	0.00154		0.00153	0.00153	
PM	0.00154	0.00000	0.00154		0.00000	
LC	0.00154	0.00000	0.00154	0.00000		

With respect to the phylogenetic trees, in the rbcL region, no nucleotide differences were observed among our sequences, which was confirmed by the grouping of the samples into a single clade in the Bayesian (100% posterior probability for the branch) tree (Fig. 6A). H. wrightii sequences from GenBank also clustered at the same node as that of our samples. For the ITS fragment, a single nucleotide difference was observed among the samples, which were split into two groups: one group with the morphology of H. beaudettei from Praia de Macapá (Piauí state) and H. wrightii from Santa Catarina and Rio de Janeiro states and a second with the morphology of H. beaudettei from Timonha-Ubatuba and H. wrightii from Pedra do Sal (also in Piauí state) (Fig. 6B), but the single base pair difference was probably attributed to the low posterior probability (PP) value of 84% for the node confidence value. Furthermore, we confirmed (with 100% confidence) separation of the Brazilian samples from the H. pinifolia and H. uninervis GenBank sequences for the two markers.

Figure 6 Phylogenetic tree for rbcL genus (A) and ITS fragment (B) by Bayesian inference parameter.

Values for posterior probability are indicated on each node. Sequences obtained from GenBank are followed by their access number and location. SETU, Timonha-Ubatuba estuarine system; PS, Praia da Pedra do Sal; MAC, Praia de Macapá; PM, Praia de Manguinhos; LC, Lagoa da Conceição.

Discussion

The present study is the first phylogenetic analysis of the genus Halodule from the Brazilian coast. Despite the significant disparity in vegetative traits across the study sites, low and no genetic divergence in the ITS and rbcL sequences was observed, respectively, for the two Brazilian morphologies, potentially hindering species discrimination that was formerly determined on the basis of only morphological criteria (den Hartog, 1967; Magalhães & Barros, 2017). These results corroborate those of Ito & Tanaka (2011), who sampled H. wrightii and H. beaudettei individuals from the northwestern Atlantic and Pacific and analyzed rbcL and phyB markers. The authors also reported insufficient genetic variation and proposed the use of the name H. wrightii s.l. This is a first step in clarifying the genetic and morphological characteristics of Halodule in Brazil; however, owing to discrepancies in the techniques, we are cautious in assuming that H. beaudettei, with an acute median tooth leaf tip, is a morphological variation in H. wrightii, with a longer lateral teeth leaf tip, versus a true species.

We found H. beaudettei only within the estuarine sites of the NB ecoregion, which is consistent with what is described by Magalhães & Barros (2017), a leaf form in the Timonha-Ubatuba Estuary that has similar leaf measures to those in our study. Halodule beaudettei meadows with these morphological characteristics were also observed along Abade Beach on Príncipe Island on the Atlantic African coast (Alexandre et al., 2017) and in two bays along the Pacific coast of Costa Rica (Samper-Villarreal et al., 2018; Samper-Villarreal, Moya-Ramírez & Cortés, 2022), with varied values of leaf length and width reported. Environmental constraints, such as hydrodynamics, sediment type and nutrients, depth, and temperature, are likely the main drivers of the variance in morphological characteristics (McMillan & Phillips, 1979; Creed, 1997; Bercovich et al., 2019; Kuo, 2020; Lin et al., 2021).

Early taxonomic studies reported that the broader and longer leaf format was confined to sheltered and muddy locations, such as estuaries, whereas the narrow and shorter leaves appeared in sandy substrates on exposed shores (McMillan & Phillips, 1979; McMillan, 1983); leaf length was assumed to grow toward greater water depths (Phillips, 1967; Bujang et al., 2008). We observed significant differences in the leaf length and width across the five sites and in the leaf length between the HB and HW populations, with the higher leaf length values displayed in the samples with H. beaudettei leaf tip shape from estuarine areas than in the other samples in this study. Since variability in plant morphological measurements was found not only between HB and HW, but also among the five sampled areas, this morphological plasticity could be related to the habitat of each population (Table S1). Thus, we recommend further studies with controlled experiments to follow the growth of these morphological characteristics; verify the role of trait variance in Halodule, such as specific leaf area, canopy height and leaf nitrogen content; and particularly, monitor the development of leaf tip forms over the life cycle of the plants in a range of habitats and environmental conditions.

Although we did not observe distinct forms cooccurring at the sampled sites in our study, previous studies reported that different leaf tip shapes grow together within the same meadow in different rhizomes and even within the same plant in Florida (Phillips, 1967; Wheeler, Furman & Hall, 2020); the results from these studies do not support the validation of H. beaudettei as a true and divergent species only by the leaf tip format (Phillips, 1967; Wheeler, Furman & Hall, 2020).

Vegetative morphology has become the main key to species identification in Halodule since reproductive organs (flowers or fruits) are rarely observed in meadows; however, it is increasingly uncertain whether distinctive features in the leaves are sufficient to sustain species separation and validation (Phillips, 1967; Wheeler, Furman & Hall, 2020). Phenotypic plasticity is a condition that allows alterations in morphological characteristics, affected by various environmental conditions, without an immediate change in the genetic code of a species. As an essential tool for adaptation to environmental and climatic changes, population monitoring can reveal how phenotypic plasticity can be accompanied by evolutionary changes and modifications in the genetic component of species. Just as the phenotypic plasticity often found in seagrasses can hide populations with high phylogenetic affinity, morphological convergence can mask genetic variation, so taxonomic assessment needs to be followed up with integrative approaches.

The haplotype network generated illustrates that the groups represented by the H. beaudettei and H. wrightii populations had the two haplotypes found in each group; thus, the two morphological variations could not be separated on the basis of their genetic variance via the ITS marker (Fig. 5). AMOVA was used to compare the genetic differences across the five populations and assigned to the groups by regions. The zero values found in the two AMOVA (Table S2) analyses indicate a lack of genetic structuring and can be explained by the high genetic similarity among the five populations and between the groups from the northeastern and southern regions. Since it was not possible to separate the groups and populations by their low genetic variation, the observed genetic variance was confined within the populations. Assessing differences between H. wrightii and H. beaudettei sequences, single nucleotide variation might explain the low values for Fst pairwise differences (Table 3), the Δst index and genetic divergence (Table 4) for the HB and HW populations and sampling sites.

Given the low genetic diversity in the sequences and clustering of the HW and HB populations in the same haplotype, the ITS marker cannot clarify the morphological variation in the Brazilian Halodule, but it could indicate that H. beaudettei and H. wrightii are closely related. The lack of genetic variation in the two markers may be driven by external factors, such as a recent founder event of Halodule on the Brazilian coast. Tavares et al. (2023) applied microsatellite markers to assess the genetic connectivity and diversity of H. wrightii across the tropical Atlantic. The authors reported low genotypic diversity in the population of Lagoa da Conceição, Santa Catarina, and high similarity among populations from northeastern Brazil and Curaçao in the Caribbean Sea. Additionally, the first record of H. wrightii in Santa Catarina state occurred in 2010 (Ferreira, 2012), suggesting a southward expansion of the species distribution (Copertino et al., 2016). The clonal growth, that prevails in H. wrightii (Rivera-Guzman et al., 2017), it might be associated with the low structure and no genetic variation found within populations.

The phylogenetic trees presented similar results, with clusters of Halodule Brazilian samples in single nodes. In the ITS tree, a second node was formed with individuals of the second haplotype but with a low confidence value (PP < 95%), resulting from small DNA sequence divergence. Additional sequences from GenBank were also grouped with our samples in the rbcL tree, indicating high similarity within samples. A sequence of H. emarginata (accession number OQ701634), downloaded from GenBank, also clustered with our samples in the ITS tree, although we could not access the specific paper to identify the morphology of this sequence.

Molecular markers selected for DNA barcoding of plants are suitable for distinguishing or merging individuals at the species level (Liu, Kumara & Hsu, 2020; Lin et al., 2021). We analyzed the plastid gene rbcL, a coding region that displays a high conservation rate, and the noncoding fragment ITS, which is hypervariable (Chase et al., 2007; Lin et al., 2021); both markers showed high similarities among all sequences, even those sampled far from each other (e.g., Piauí and Santa Catarina, located at a distance of 4,800 km). Previous phylogenetic analyses involving the sequencing of the rbcL region yielded contrasting results, with successful species delineation when complementary molecular markers were used (Kato et al., 2003; Kress & Erickson, 2007; Lin et al., 2021), but they could also not reveal relevant trends in the genetic relationships of seagrass species (Lusana & Lugendo, 2023). The latter was also visualized in this research for Halodule in Brazil, most likely as a result of the low-resolution performance of the rbcL gene; thus, it is necessary to expand the analysis with new markers with better resolution. Additional markers, such as the trnH-psbA intergenic spacer region or preferably the entire genome, should be evaluated (Bricker et al., 2011) to provide more insights into genetic distances in Halodule.

Whole-genome sequencing can reveal whether there is genetic variation to separate the HB and HW populations, since the low genetic divergence in the ITS marker found to date emphasizes the phylogenetic closeness between the two Halodule morphologies; however, we found significant morphological distinctions, in terms of not only leaf tip but also leaf length. The consistency between the results of the morphological and genetic analyses highlights the taxonomic challenges in seagrasses and the importance of caution in the delimitation of species status. Morphological plasticity is a potential driver of phenotypic diversity in leaves and H. beaudettei could still be considered a morphological variation of H. wrightii. We recommend that a genomic approach should be used to solve this incongruence within Halodule genus in Brazil.

Conclusions

There is a large gap in ecological studies of Brazilian seagrass meadows. Our results suggest insufficient genetic variation in the ITS and rbcL markers to discriminate between H. wrightii and the samples with H. beaudettei morphology; however, some morphological measurements were distinct between the two forms and among the sampling sites. Morphological plasticity can be the main driver of differences in leaf characteristics. The resulting low haplotypic diversity and genetic distance between HB and HW and across the five populations are indicative of possible demographic effects, such as clonal growth and recent colonization events. However, this study also highlights the phylogenetic closeness between H. wrightii and H. beaudettei, at least with respect to the ITS marker, largely used for DNA barcoding. Thus, we emphasize the need to combine morphological and genetic analyses—preferably genomic markers—to resolve taxonomic issues in Halodule and adequately understand the distribution patterns and range of each species to enhance management plans for the conservation and restoration of seagrasses. The phylogenetic relationships within the genus will be better resolved with comprehensive sampling across its distribution in the Caribbean and eastern Atlantic and Pacific coasts, as well as with additional sampling of H. emarginata individuals to confirm or reject H. wrightii as a monospecific clade in the Atlantic and eastern Pacific.

Supplemental Information

Supplemental Information 1 Sampling and sites information.

Supplemental Information 2 AMOVA results for the five populations and the groups formed by the northeast (PS, MAC and SETU) and south (PM and LC) regions.

Supplemental Information 3 R code for morphological analysis.

Supplemental Information 4 CTAB Protocol.

Modifications in this study.

Supplemental Information 5 Leaves morphology measures for all the sites.

SETU: Timonha-Ubatuba estuarine system; PS: Praia da Pedra do Sal; MAC: Praia de Macapá; PM: Praia de Manguinhos; LC: Lagoa da Conceição.

We thank Paulo Horta, Carlos Eduardo Dias and Hanna Brum for their help with sampling surveys and laboratory assistance. We appreciate and express our gratitude to Felipe Dumont and Juliana de Biasi, for comments and suggestions on the manuscript.

Additional Information and Declarations

Competing Interests

The authors declare that they have no competing interests.

Author Contributions

Virgínia Eduarda de Sousa conceived and designed the experiments, performed the experiments, analyzed the data, prepared figures and/or tables, authored or reviewed drafts of the article, and approved the final draft.

Maria Cristina F. da Silva Cortinhas conceived and designed the experiments, performed the experiments, analyzed the data, prepared figures and/or tables, authored or reviewed drafts of the article, and approved the final draft.

Joel C. Creed conceived and designed the experiments, authored or reviewed drafts of the article, and approved the final draft.

Maria Gardênia S. Batista conceived and designed the experiments, performed the experiments, prepared figures and/or tables, and approved the final draft.

Maira C. Proietti conceived and designed the experiments, analyzed the data, authored or reviewed drafts of the article, and approved the final draft.

Margareth Copertino conceived and designed the experiments, authored or reviewed drafts of the article, and approved the final draft.

Field Study Permissions

The following information was supplied relating to field study approvals (i.e., approving body and any reference numbers):

Field collection were approved by the Sistema de Autorização e Informação em Biodiversidade (SISBIO) (License number: 45819).

DNA Deposition

The following information was supplied regarding the deposition of DNA sequences:

The group ITS and rbcL sequences are available at GenBank: OR284885 to OR284889 and OR345358 to OR345362, respectively.

Data Availability

The following information was supplied regarding data availability:

The data is available in the Supplemental File.

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
