# Peer review of "Assessing morphological variations in the seagrass genus Halodule (Cymodoceaceae) along the Brazilian coast through genetic analyses"

_PeerJ, doi:10.7717/peerj.19038_

## Round 0.1 · original submission · Major Revisions

Thank you very much for submission of an interesting manuscript to Peer J. All three reviewers basically agreed to accept the manuscript that is well written. However, extensive revision is needed before making final decision, because reviewers commented requirement of a lot of corrections. Please revise the manuscript according to reviewer’s comments.

Reviewer 1 ·

Basic reporting

Sousa et al. demonstrated a long-lasting issue among Halodule related to the ability for morphological plasticity that confuses the taxonomic works. They incorporated molecular and traditional approaches to address this issue. In general, the writing has significant room to improve and may need to be checked by a fluent English speaker to ensure the quality of English meets the standard for publication. The primary issue of this manuscript is their hypothesis, in the content, the taxonomic status of H. wrightii and H. beaudettei is inconsistent. Based on the literature and WoRMS system, H. beaudettei is a synonym of H. wrightii. And across the content, they called it a morphotype in some sentences but sometimes called it putative H. beaudettei. Even in the title they seem to treat it as a valid species which greatly confused me. I suggest the authors reframe this manuscript's structure to clarify their goal.

Experimental design

There are two thing that I am concerned, The first is the sample size of this study, in total, they obtained 30 plants for genetic analyses but did not clearly state how many were at each site for each morph. In the morphological examination, 50 samples were used but did not provide how many were at each site for each morph. Are these two sets of samples corresponding or have they been collected separately? I hope they can address this missing info with a table and its corresponding GenBank accession No. The second, without examining the voucher of H. emarginata, it is risky to assume it is closely related to wrightii. Since Halodule is a genus notorious for morphological plasticity, caution is needed to assume this.

Validity of the findings

The findings of this study are concordant with previous findings of Ito and Tanaka (2011), therefore, the impact and novelty are not very high in terms of general findings.
Due to their small sample size, a particular bias may occur. However, the phylogenetic analyses they conducted followed a standard process.
As I suggested, they may need to reframe the structure to form a more logistic flow.

Additional comments

line 22-24: Please rephrase this sentence, it's unclear what you are trying to express.
line 59: What is XX means?
Line 61-63: Rephrase is needed.
line 90-92:They didn't use any molecular approach. Based on the results of this study, H. beaudettei has been identified as a synonymy of H. wrightii. So, I suggest you to reframe your manuscript towards may be "molecular identification of Halodule species in the Brazilian coastal water ". And in Wheeler et al. 2020, they mentioned the incongruent results of the phylogenetic relationship between these two species in the literature, you should use this as the hypothesis to test with a broader sampling scheme and morphological quantification.
line95-97:Since H. beaudettei has been listed as a synonym, it could be the reason why the current status of it is listed as data deficient. As I mentioned in a previous comment, you can not treat H. wrightii and beaudettei as two valid species.
line109-112: You should include the papers published by
Ito and Tanaka 2011, Wagey and Calumpong, 2013
to address the consistency of molecular works on Halodule.
line 137: I do not think using "hypothetical species" is appropriate.
line 138: "A minimum of 10 plant specimens were collected in each site" for molecular or morphological analysis? Could you make it clear?
line 158:I strongly suggest the authors add a tale that shows the info of sites, the sample size of each site, and the corresponding morphological trait (tip morph). Which could help a lot to readers.
line 172:Lucas, Thangaradjou & Papenbrock, 2015, "respectively",
line 178: replace "submitted" to "running"
line 202: What does "newly inserted" mean? Did you mean "newly submitted"? Rephrasing the sentence is needed.
line213-219: The results of morphological analyses should be demonstrated with a bar chart to visualize the differences.
line 223: How about the results of H. beaudettei /another morph of wrightii?
line 241:You used different terms to refer to the species you worked on including potential species and morphotypes. However, they have different definitions, in your title, you seem to treat them as two valid species, but here you called them as morphotypes? Based on the literature, you should treat them as two morphotypes of H. wrightii and test whether if you can find a genetic variation between them. Again reframing the manuscript is essential to get it published.
line 261:Another example, you treated it as a valid species here.
line 280: use "hyper variable" instead of "a more polymorphic site"
line 293, please rephrase
line 294-295: This sentence is self-contradiction. When you call it a neutral marker, it refers to selection-free... You have to work on this paragraph since it's not very clear.
line304: again you treated both species as valid species here.
line305-307: A genomic approach is needed to solve this issue because ITS and RBCL may not provide enough resolution for recently derived species pair. Therefore, you have to integrate the morphological examination with a genomic approach instead of the method that you are using now.
line313-315: This sentence is confusing. You have to rewrite it to make it clear. Plasticity is one kind of mechanism casing morphological variation. The transcriptomic analysis could reveal it. You should not mix it with the morphological species concept that taxonomists generally apply.

Reviewer 2 ·

Basic reporting

The manuscript explains that based on morphology and rbcL and ITS sequences Halodule wrightii and H. beaudettei (Cymodoceaceae) are not distinct species in the Brazilian coast.

The manuscript is well written. The seagrass species in Halodule is known to have taxonomic complexes, due to morphological plasticity. In this manuscript the authors used rbcL and ITS sequences to analysed the differences between H. wrightii and H. beaudettei.

Experimental design

Among other chloroplast sequences, why rbcL was used? matK is considered a marker with the highest species discrimination for plant identification (Cowan and Fay 2012). Also any reason not to try mitochondrial sequences?

Validity of the findings

In results section of Leaf Morphology please describe the characteristics of leaf tip of H. beaudettei and H. wrightii or the description of the two species. How they are differ morphologically. The morphological data of each sample may be showed in a table

Discussion:
lane 243: may need other markers to further conclude that they are same species
lane 245: what cause this different morphology between the two sp
Do environmental conditions induce the morphotype of H. beaudettei? How are the conditions at NB ecoregion? Since it is only found in this area. Is environmental condition at NB same as other location whre H beaudettei found?
What about hybridization? Is it possible that there are interspecific hybridization in Halodule that results in new species?

Figure 2. Which one are H. wrightii? and Which one are the putative H. beaudettei ?

Additional comments

no additional comments

Reviewer 3 ·

Basic reporting

Please refer to the PDF attachment for all my comments

Experimental design

Please refer to the PDF attachment for all my comments

Validity of the findings

Please refer to the PDF attachment for all my comments

Additional comments

Please refer to the PDF attachment for all my comments

Annotated reviews are not available for download in order to protect the identity of reviewers who chose to remain anonymous.

---

## Round 0.2 · Major Revisions

Two reviewers basically agree to accept the manuscript; however, many concerns were still mentioned. Revise the manuscript carefully according to all comments from the reviewers for improvement.

Reviewer 1 ·

Basic reporting

The paper entitled "Assessing morphological variations in the seagrass genus Halodule (Cymodoceaceae) through genetic analyses in the Brazilian coast" by Sousa et al. which mainly uses morphological and molecular approaches to reveal the species boundary among Halodule on the Brazilian coast. First of all, thank you for submitting your revision with the consideration of comments from reviewers. There are a few things that I think you have to take care of before moving to the next stage. You took my advice to reframe your manuscript but you didn't define the species status of Halodule wrightii and H. beaudettei very well across your manuscript. I suggest you consider H. wrightii as a valid species and H. beaudettei as its morphological form, so describe them as H. wrightii and H. beaudettei form through out the manuscript including the usage in the figures and table. And there are numerous places that you used "," to replace ".", please double check this issue before you send out the next revision. I have included all my comments in the PDF file, I look forward to reading your next revision.

Experimental design

no comment

Validity of the findings

no comment

Additional comments

All my comments are enclosed in the attached PDF file.

Annotated reviews are not available for download in order to protect the identity of reviewers who chose to remain anonymous.

Reviewer 3 ·

Basic reporting

It is still unclear why the rbcL and ITS molecular markers were chosen over other widely used markers such as matK, trnH-psbA and etc..

Experimental design

1) The R package used for the ANOVA analysis should be specified in the analysis section. Additionally, the R scripts used for this analysis should be made available.
2) As per the authors' response, “the sequences were manually aligned in BioEdit”. However, this approach requires further clarification. While manual alignment can be acceptable in some cases, it may introduce subjective bias if not conducted with clear guidelines. Could the authors provide more details on the specific criteria or procedures they followed during this manual alignment? Additionally, manual alignment is generally considered less reliable compared to algorithmic methods. Were any validation steps or checks implemented to ensure the accuracy and reliability of the alignments?
3) What specific parameters or settings were used in AMOVA? Can these be detailed?
4) How were the 1,000,000 generations chosen for the BI analysis? Were any convergence diagnostics performed to ensure that this number of generations was sufficient, and if so, could you provide details on how convergence was assessed?
5) In your explanation, you state that "Each analysis considered the most suitable evolutionary models, as defined in JModelTest 2.1.9: HKY+G for ITS and K80+I for rbcL for both BI and ML methods." Could you provide more details on the criteria used to determine that these models were the best fit for your data?

Validity of the findings

1) The limited genetic diversity observed with the rbcL marker raises the earlier question of why this marker was chosen over other widely used markers, especially considering that previous researches have demonstrated its unsuitability for species separation within the genus Halodule.
2) The diversity results for the ITS region show low haplotype diversity (h = 0.4945) and nucleotide diversity (π = 0.0007) in the Piauí population, while no haplotype or nucleotide diversity was observed in the Santa Catarina and Rio de Janeiro populations. Despite this, the AMOVA indicated significant differentiation among populations (Fst = 0.55818). Could the authors clarify how such low overall genetic diversities correspond to the significant Fst value observed? Additionally, the specific parameters or settings used for the AMOVA analysis were not provided.
3) The explanation for figure 5 does not fully acknowledge the uncertainty in this grouping. While the authors report the split between the two morphologically distinct groups, the low posterior probability and bootstrap indicate that this genetic differentiation might not be robust enough to draw definitive conclusions. The authors should be more cautious when interpreting this result, as the genetic evidence is not strongly conclusive.

Additional comments

1) The authors assert that the lack of genetic variation corroborates previous studies, but the discussion does not sufficiently address the limitations of these markers in capturing species-level differences.
2) The discussion acknowledges high Fst values in the AMOVA results for the ITS region (Fst > 0.25, indicating significant population differentiation). Yet, this high genetic differentiation is downplayed, with the authors suggesting that the low number of haplotypes and polymorphic sites might mask the "real diversity" between populations. This creates a contradiction- the genetic differentiation is significant according to the Fst values (0.55818), but the authors argue it is not sufficient to support species distinction. The authors should clarify this apparent contradiction. Instead of dismissing the significant Fst values, they could explore whether the observed genetic differentiation is indicative of incipient speciation or population structure.
3) While the rbcL marker is indeed widely used for DNA barcoding of plants but its high conservation rate makes it less suitable for resolving relationships at the species level in certain groups, including seagrasses like Halodule. Studies by Lusana and Lugendo (2023) and Stevanus and Pharmawati (2021) have demonstrated that rbcL lacks sufficient interspecific variation to differentiate closely related species within Halodule. Similarly, Lucas et al. (2012) showed that rbcL can only resolve species down to the genus level in Halodule. Given the well-documented limitations of rbcL in Halodule, the justification for using this marker in your study is not adequately provided. The absence of genetic divergence observed in your study is likely a result of the marker's low resolution, rather than an accurate reflection of conspecificity.
4) Insufficient discussion on morphological plasticity. While the authors mention the high morphological variability in seagrasses and how environmental factors could influence this, they do not fully explore the possibility that H. beaudettei's distinct morphology may result from environmental plasticity rather than conspecificity with H. wrightii. They attribute the morphological differences to environmental constraints but fail to critically analyze whether this could mask underlying genetic divergence.
5) The authors briefly mention the possibility of an ongoing speciation process but do not explore this hypothesis in sufficient depth. They quickly dismiss it in favor of morphological plasticity without providing enough evidence for either scenario. The discussion around this potential speciation event is superficial and seems to contradict the earlier assertion that H. beaudettei is a morphological variant.

---

## Round 0.3 · Major Revisions

The current version of the manuscript is requested improvement of the language quality by the Reviewer 1 and revision of minor points by the reviewer 3. The authors must respond to these requests in the next version of the manuscript.

Reviewer 1 ·

Basic reporting

The authors have revised their manuscript according to our comments, however, the latest revision is not easy to read. I have tried my best to help but there are many sentences that I have no idea what they are trying to express. I strongly suggest the authors send their manuscript for English polishing before the next submission. Since English polishing and proofreading are not our responsibility. If this is not been improved for the next round I may reject your paper without further review.

Experimental design

No comment

Validity of the findings

It's not novel, but it's legit.

Additional comments

All my comments and corrections are included in the attached file.

Annotated reviews are not available for download in order to protect the identity of reviewers who chose to remain anonymous.

Reviewer 3 ·

Basic reporting

NA

Experimental design

NA

Validity of the findings

NA

Additional comments

1. Line 117-119: The sentence seems out of place and lacks a clear connection to the rest of the paragraph.
2. Line 126: Change “will be” to “was”
3. Replace +/- with ± in table 2.
4. Line 315-317: With the low/no genetic diversity observed, how do you support the claim that genetic diversity may explain the observed leaf tip variability?
5. Line 331-333: Does 84% support separating H. beaudettei and H. wrightii not suggest meaningful genetic differentiation among these species?
6. Line 338 -343: How do you reconcile the finding of 0.00 genetic variance among populations with your statement that there is “high genetic structuring between populations”? Does the observed 100% variation within populations not suggest that any genetic structuring is confined to within-population differences rather than between-population differentiation?
7. Line 343: The phrase “Contrasting results” is misleading, as the results mentioned above are not contradictory. The authors failed to correctly interpret the results.
8. How do you explain the AMOVA showing 100% variation within populations (no genetic structuring among populations) and pairwise FST values of 1.000 between population (complete genetic structuring between populations)?
9. Was the observed polymorphic site in your analysis parsimony informative or a singleton?
10.The expanded discussion on morphological plasticity is an improvement, but it still feels somewhat cursory. The authors mention environmental influences on morphology but do not critically engage with whether such plasticity could be masking underlying genetic variation or contribute to phenotypic convergence.

---

## Round 0.4 · Major Revisions

Please revise according to comments from the reviewer 1. Main concern is interpretation of the ANOVA and FTS data.

Reviewer 1 ·

Basic reporting

The writing has improved a lot. However, I have made several corrections in the PDF file that I attached to help them polish it. I am satisfied with the data they provided on the morphological and phylogenetic analyses but not the data they provided on population genetic analyses i.e. AMOVA and FST. I think the authors may need to take a close look at their data. In the new version they re-ran the FST and results changed from "1" to "0", and they did not mention what setting they changed in the new analysis in the manuscript which is disturbing. In the reply to the question 8 I quoted here "8. How do you explain the AMOVA showing 100% variation within populations (no genetic structuring among populations) and pairwise FST values of 1.000 between population (complete genetic structuring between populations)?
We re-ran the pairwise Fst calculations between the populations, but this time assuming the distance between the haplotypes, and given the low haplotypic diversity, we obtained new values of 0.00. This indicates that there is little genetic difference between the populations.
Whereas the value of 100% variance within populations, we consider to be the result of low sampling and the absence of genetic variation between individuals, since there is no variation between the groups and populations. We therefore question the robustness of an AMOVA analysis."
Based on their reply, I can't understand their logic and doubt they know how to interpret the results of AMOVA and FST correctly. In your case, you have only two haplotypes and these two haplotypes are not occurring in the same location. When you look at the new figure (Figure 5) that I requested. PM and LC have only haplotype 2 and you can say there is high gene flow between them (FST=0), but no geneflow (FST=1) between PM, LC and PS, SETU since PS and SETU have only haplotype 1. And at the end of your reply, you mentioned "since there is no variation between the groups and populations. We therefore question the robustness of an AMOVA analysis." Are you implying you included the questionable results in this manuscript ?? This is the last opportunity that you can fix it, if you can not give a clear explanation in your reply as well as in the manuscript in the next run, I will consider rejecting this paper.

Experimental design

for the AMOVA grouping, group by two different haplotypes does not make sense. Normally, you can group them based on population (5 populations) or geographic location(northern and southern).

Validity of the findings

except the population genetic part, rest of them are ok to me.

Additional comments

Table 4 is also a mess please check my comment in the PDF file.
Line 239, you didnt present the genetic divergence values estimated by MEGA.

Annotated reviews are not available for download in order to protect the identity of reviewers who chose to remain anonymous.

Reviewer 3 ·

Basic reporting

NA

Experimental design

NA

Validity of the findings

NA

Additional comments

NA

---

## Round 0.5 · accepted · Accept

The novel revised version was evaluated positively. Thus, the manuscript is now suitable for publication in PeerJ.

Reviewer 1 ·

Basic reporting

Congratulation! I am happy with your replies and corrections. I know it's not a smooth way to get through the revision process. But I hope you can learn how to be well-prepared for your future manuscript. Again, congrats!

Experimental design

all good.

Validity of the findings

all good.